# Environmental signal shredding on sandy coastlines

Eli D. Lazarus[1]*, Mitchell D. Harley[2], Chris E. Blenkinsopp[3] and Ian L. Turner[2]

[1]Environmental Dynamics Lab, School of Geography and Environment, University of Southampton, Southampton, UK

[2]Water Research Laboratory, School of Civil and Environmental Engineering, University of New South Wales, Sydney NSW, Australia

[3]Research Unit for Water, Environment and Infrastructure Resilience (WEIR), University of Bath, Bath, UK

*correspondence to: E.D.Lazarus@soton.ac.uk

**ORCIDs**

| | |
|---|---|
| Lazarus | 0000-0003-2404-9661 |
| Harley | 0000-0002-1329-7945 |
| Blenkinsopp | 0000-0001-5784-2805 |
| Turner | 0000-0001-9884-6917 |

**Abstract**

How storm events contribute to long-term shoreline change over decades to centuries remains an open question in coastal research. Sand and gravel coasts exhibit remarkable resilience to event-driven disturbances, and, in settings where sea level is rising, shorelines retain almost no detailed information about their own past positions. Here, we use a high-frequency, multi-decadal observational record of shoreline position to demonstrate quantitative indications of morphodynamic turbulence – "signal shredding" – in a sandy beach system. We find that, much as in other dynamic sedimentary systems, processes of sediment transport that affect shoreline position at relatively short time-scales may obscure or erase evidence of external forcing. This suggests that the physical effects of annual (or intra-annual) forcing events, including major storms, may convey less about the dynamics of long-term shoreline change – and vice versa – than coastal researchers might wish.

**Keywords –** coastal hazard; landscape resilience; beach recovery; beach rotation; Narrabeen-Collaroy

## 1. Introduction

Quantifying magnitudes and rates of shoreline change is fundamental to understanding the dynamics of coastlines: not only how they behave over time, but also how they may respond to future changes in environmental forcing. From a coastal-management perspective, shoreline change may constitute a coastal hazard – either event-driven, like the impact of a major storm, or chronic, like persistent shoreline erosion from a net-negative sediment budget. Long-term, continuous measurement of shoreline position observed at a given location will record changes arising from event-driven and chronic forcing, alike. But how punctuated storm events contribute to long-term shoreline change over decades to centuries remains an open question, particularly in the context of shoreline-change prediction (Morton et al., 1994; Fenster et al., 2001; Houser and Hamilton, 2009; Anderson et al., 2010; Masselink and van Heteren, 2014; Brooks et al., 2016; Masselink et al., 2016; Scott et al., 2016; Burvingt et al., 2017).

Evidence of coastal storm frequency and magnitude over centuries to millennia may be stored in the sedimentary stratigraphy of beach ridges (Tamura, 2012) and washover into back-barrier lagoons (Donnelly and Woodruff, 2007). Ridge and washover stratigraphy offers a window into climatic forcing conditions in the recent geologic past, but is not a direct measure of shoreline position. Indeed, in transgressive settings (in which relative sea level is rising) the shoreline itself retains almost no detailed information about its own past positions. Sand and gravel coastlines, especially, reflect remarkable resilience to event-driven disturbances – even to tsunami (Choowong et al., 2009). Storm-driven shoreline excursions on the order of $\sim 10^1$–$10^2$ m may be obscured within days to months, and effectively erased within years (Birkemeir, 1979; Egense, 1989; Thom and Hall, 1991; Morton et al., 1994; Douglas and Crowell, 2000; Honeycutt et al., 2001; Zhang et al., 2002; List et al., 2006; Lazarus et al., 2012; Lentz et al., 2013; Coco et al., 2014; Masselink and van Heteren, 2014; Phillips et al., 2017).

This coastal context exemplifies a unifying challenge in geomorphology: determining how dynamic sedimentary systems – especially source-to-sink pathways – respond to rapid external forcing. Processes of sediment transport tend to rework upstream/upslope inputs so completely that their downstream/downslope outputs may bear no resemblance to the original pattern of forcing that drove them. In their essential synthesis of the problem, Jerolmack and Paola (2010) call this phenomenon the "shredding" of environmental signals. They offer that shredding – or, more formally, "morphodynamic turbulence" – behaves much like fluid turbulence, in that "energy injected at one frequency is smeared across a range of scales." High-frequency signals of external forcing are especially likely to be shredded. Drawing on the physics of turbulent fluid flows (Frisch and Kolmogorov, 1995), Jerolmack and Paola (2010) used time-series of sediment flux from physical and numerical experiments – bedload transport in a flume channel (Singh et al., 2009), a canonical rice-pile experiment (Frette et al., 1996), and a numerical rice-pile model – to illustrate their argument. Beyond source-to-sink sedimentary systems (Romans et al., 2016), signal shredding has since been extended to spatio-temporal changes in lake levels (Williams and Pelletier, 2015) and methane release from peatlands (Ramirez et al., 2015).

Here, we investigate signal shredding in an altogether different sediment-transport system:
that of a sandy beach. Although previous studies of sandy shoreline dynamics have invoked
signal shredding conceptually (Lazarus et al., 2011a, 2012; Williams et al., 2013), none have
used observations of shoreline position to demonstrate quantitative signatures of signal
shredding empirically. Following Jerolmack and Paola (2010), we find the hallmarks of
morphodynamic turbulence in time-series of shoreline position measured at Narrabeen-
Collaroy Beach, in southeast Australia (Short and Trembanis, 2004; Harley et al. 2011a,
2015; Turner et al. 2016; Phillips et al., 2017). The potential for beaches to "shred" large-
magnitude changes in shoreline position forced at relatively short (~intra-annual) time-
scales complicates reconciliation of short-term beach dynamics and long-term, spatio-
temporal patterns of shoreline variability and evolution.

## 2. Setting and datasets

The Narrabeen-Collaroy embayment (Fig. 1a) holds a sandy beach 3.6 km long, and is one
of only a few sites worldwide where ongoing beach monitoring has been regular, frequent,
and uninterrupted for multiple decades (Turner et al., 2016). Cross-shore profiles at five
locations along the beach (Fig. 1a) have been measured approximately monthly (Fig. 1b)
since 1976 (Turner et al., 2016). In addition, continuous alongshore shoreline positions
derived from RTK-GPS quad-bike surveys of the full three-dimensional subaerial beach
have been recorded approximately monthly (Fig. 1c) between 2005–2017 (Harley and
Turner 2008; Harley et al., 2011a, 2011b, 2015). Daily-averaged shoreline position in the
southern half of the embayment (Fig. 1a) has also been captured by an Argus Coastal
Imaging system (Fig. 1d) for over a decade (Phillips et al., 2017). In each of these datasets
we used the 0.7 m AHD (Australian Height Datum) elevation contour to define the cross-
shore shoreline position ($x$) at all positions alongshore ($y$), commensurate with mean high
water (Harley et al., 2011a, 2011b). Data gaps in the profiles and time-series were filled by
linear interpolation. We also used deep-water wave data compiled from hourly records
logged between 2005–2017 by the Sydney waverider buoy, located approximately 11 km
offshore of the study area.

## 3. Analysis

### 3.1. Patterns in power spectra

In their bedload and rice-pile examples, Jerolmack and Paola (2010) collapsed these
physical systems into one dimension – a time-series of sediment flux past a single point. In
our beach example, rather than considering sediment flux directly, we tracked the change in
shoreline position, $d_x$ (in m), between consecutive time steps at a given position alongshore
($y$). In a generic source-to-sink system in which sediment only moves downstream,
sediment flux is unidirectional and positive. By contrast, in a one-dimensional treatment of
a beach system, shoreline movement ($d_x$) is bidirectional, as wave-driven cross-shore
sediment transport shifts the shoreline at any location onshore and offshore over time. To
therefore include both onshore (negative) and offshore (positive) movement, we worked
with the absolute value of shoreline change and calculated the power spectrum of the time-
series using wavelet analysis, following the method described by Lazarus et al. (2011a,
2012). We show results based on the median absolute value of shoreline change for all
positions alongshore at a given time step (Fig. 2a–c). To confirm that this simplification is
representative, we also analysed the spectral density of the shoreline-change time-series at
each position alongshore (Fig. S1).
This application of wavelet analysis functions much like a Fourier transform (Lazarus et al.,
2011a, 2012). We first convolved the time-series (the absolute value of shoreline change)
with a second-order Daubechies wavelet in a continuous wavelet transform. Taking the
mean transform variance at temporal scales up to approximately half the overall length of
the signal produced a measure of spectral power. We chose a wavelet with a small number
of vanishing moments – a measure of how much the wavelet shape undulates – because
simple wavelets tend to have better sensitivity over a greater range of scales. The general
pattern of spectral density was insensitive to different wavelets with low vanishing
moments, and was comparable to spectra generated by a Fast Fourier Transform (Fig. S2).
Like the sedimentary systems described by Jerolmack and Paola (2010), the spectral density
of the one-dimensional shoreline-change term $d_x(t)$ yields a pattern with two regimes (Fig.
2d). A non-stationary regime extends over shorter time-scales, such that spectral density
versus time-scale are correlated by a power law. This relationship transitions at ~9–11 mos
into a comparatively stationary (uncorrelated) regime over longer intervals. (A power
function fitted to the three spectra, combined, for scales up to ~12 mos, returns a scaling
exponent = 0.66, but the physical significance this slope value remains unclear.) This two-
regime pattern in the power spectrum (Jerolmack and Paola, 2010) serves as an initial
indication that signal shredding may be inherent in the dynamics of sandy beach systems.
But what environmental signal is being shredded at the shoreline? Consider again a
unidirectional source-to-sink system, driven by some input flux at the upstream end. That
input flux might be constant; it might fluctuate quasi-periodically; it might spike with large-
magnitude events. In a controlled physical experiment or a numerical model, input flux (of
sediment and/or fluid) is a known quantity, set by the researcher. Whatever its pattern in
time, input flux embodies the environmental signal that is susceptible to shredding by
sediment-transport processes internal to the system. Here, for the beach system, we treated
energy flux from incident storm waves as the external environmental signal that shoreline
behaviour may destroy or preserve.
Previous work on Narrabeen-Collaroy has demonstrated that the relationship between
wave-energy flux and shoreline change is strongest for storm waves (Harley et al., 2009;
Phillips et al., 2017). By isolating storm waves, we do not mean to suggest that lower-
energy waves do not move sediment. However, changes in nearshore bar and beach
morphology tend to emerge far more slowly than the high-frequency variability of low-
energy wave forcing (Plant et al., 2006), and, in this case, we are interested in the conditions
under which an input flux could be preserved in the shoreline response signal. We defined
storm wave conditions by a threshold corresponding to the 95th percentile of deep-water
significant wave height ($H_s$, m), which for this region is $H_s > 3$ m (Harley, 2017). Much like
flow discharge in a fluvial system, deep-water wave energy flux ($E$, kW per m wavefront)
may serve as a useful proxy for input flux to the beach:

$$E = \frac{\rho g^2}{64\pi} H_s^2 P_w \approx 0.5 H_s^2 P_w \tag{1}$$

where $\rho$ (kg/m$^3$) is water density, $g$ (m/s$^2$) is acceleration by gravity, $H_s$ (m) is significant
deep-water wave height, and $P_w$ (s) is wave period (Herbich, 2000).
We calculated monthly and daily total storm-wave energy fluxes corresponding to the
monthly and daily shoreline time-series (Fig. 2e,f), and transformed them into power
spectra to demonstrate that the forcing (input) and response (output) spectra are not the
same (Fig. 2d,g). Where the spectral density of shoreline change is non-stationary
(correlated) over a range of relatively short time-scales (Fig. 2d), the spectral density of
wave forcing is comparatively stationary (uncorrelated) over the same range (Fig. 2g). The
monthly wave-energy time-series shows a peak in spectral density at ~24 mos, but with no
clear comparator in the shoreline-change spectra. The daily wave-energy spectrum rises at
the long-interval end of its range to a broad peak at ~30–45 mos (Fig. 2g), which overlaps
with a local maximum in the shoreline-change spectra at ~37–42 mos (Fig. 2d).
Even in this one-dimensional representation, the sediment-transport processes of shoreline
change have transformed an input signal into a quantitatively distinct output signal. To
place these input/output spectral patterns in the context of physical processes that might
explain them, we explored characteristic time-scales of key embayed-beach dynamics.

**3.2 Characteristic time-scale from system size and input flux**

Jerolmack and Paola (2010) showed in their exemplars that the transition from non-
stationary to stationary (correlated to uncorrelated) in the spectral density of the output
signal occurs at an intrinsic, characteristic time-scale $T_c$. Theoretically, $T_c$ is set by the
system size $L$ relative to the constant (~mean) signal input. While those parameters can be
dictated for experimental systems, they are less clear for an open sandy coastline. To
independently estimate $T_c$ in the Narrabeen-Collaroy system and compare the results to the
time-scale (or range of time-scales) at which the shoreline-change power spectra transition
from non-stationary to stationary, we tested two different approaches.
The first approach is a back-of-the-envelope exercise. We assumed that the system size $L$ is
equivalent to maximum cross-shore beach width, defined here as the cross-shore distance
from a fixed landward reference point to mean sea level (Harley and Turner, 2008; Harley
et al., 2011b). This assumption extends from having collapsed the system into only the
cross-shore ($x$) dimension: at any alongshore position ($y$), the theoretical maximum cross-
shore ($x$) extent to which the beach can ever erode is the full width of the beach $L$,
independent of embayment length. (We call $L$ the "theoretical maximum" because
historical records of shoreline change are necessarily of finite duration, and therefore may
never reflect this full width.) We normalised $L$ relative to its maximum value, such that the
theoretical maximum $L = 1$. For the input flux, we took the mean normalised monthly
(and daily) total wave-energy flux over the full span of the dataset, which here serves the
purpose for a rough estimate of $T_c$. Using monthly total storm-wave energy flux (Fig. 2e),
$L/E$ (where $L$ and $E$ are both normalised) yields $T_c$ = 4–6 months; using the daily total
storm-wave energy flux (Fig. 2f), $T_c$ = 5–6 months. (These ranges come from excluding
and including, respectively, zero values in the total wave-energy time-series, which increases
or decreases the mean normalised $E$.) Note that this estimate aligns with a detailed analysis
of time-scales for beach recovery at Narrabeen-Collaroy (Phillips et al., 2017). Plotted in
relation to the power spectra for shoreline change (Fig. 2d), the characteristic time-scale
marks approximately where the spectral density "rolls over" from non-stationary to
stationary (correlated to uncorrelated), just ahead of the distinct local maximum at ~9–11
months.

### 3.3 Characteristic time-scale from modes of embayed beach dynamics

The second approach to estimate one or more characteristic time-scales $T_c$ for the
Narrabeen-Collaroy system derives from shoreline behaviours typical of this site, and of
embayed beaches more generally (Short and Trembanis, 2004; Ranasinghe et al., 2004;
Harley et al., 2011a, 2015; Ratliff and Murray, 2014).
Although they vary in detail between specific locations, approximately four modes of
shoreline behaviour tend to describe how sediment moves within embayed beach systems.
One mode represents sediment cycling offshore and onshore as a quasi-coherent unit at
the full scale of the embayment: imagine a narrow beach during stormier times of the year,
and a wide beach during calmer intervals. Another common mode is termed "rotation,"
and occurs when prevailing wave conditions or a storm event shifts a significant volume of
sediment inside the embayment alongshore to form a wider beach at one end and a
narrower beach at the other (Ranasinghe et al., 2004). Related to rotation is what has been
described as a "breathing" mode, a kind of shoreline resonance that hinges near the centre
of the beach and characterises changes in shoreline curvature, as sand moves between the
middle and ends of an embayment (Ratliff and Murray, 2014). An additional mode of
shoreline dynamics reflects patterns of shoreline variability introduced by rhythmic
movements of sandbars, sandwaves, megacusps, and inlet processes, where applicable
(Harley et al., 2011a, 2015). These four modes are not necessarily hierarchical: their relative
dominance can change as a function of wave conditions (Harley et al., 2011a, 2015). More
importantly, these modes of shoreline behaviour likely manifest intrinsic time-scales.
To find characteristic time-scales corresponding to the modes of shoreline behaviour at
Narrabeen-Collaroy, we followed steps described by Ratliff and Murray (2014). From the
monthly shorelines derived from RTK-GPS quad-bike surveys, at each position alongshore
we detrended the series of shoreline position (not shoreline-position change) in time (Fig.
3a). To calculate the empirical orthogonal modes in the alongshore dimension through
time, and thus characterise shoreline variation around its mean position (Fig. 3b), we
applied principal-component analysis (Winant et al., 1975; Aubrey, 1979; Clarke and Eliot,
1982; Hsu et al., 1994; Dail et al., 2000; Short and Trembanis, 2004). Each mode in
sequence explains a smaller percentage of variation in the data. We then used a continuous
wavelet transform, again finding the mean transform variance over a range of time intervals
(Lazarus et al., 2011a), to examine the spectral signatures of the first four behavioural
modes in the temporal dimension. In the resulting power spectrum, peaks represent the
characteristic time-scale for each behavioural mode (Ratliff and Murray, 2014). We take $T_c$
(Fig. 3c) as the first local maximum in the power spectrum (Ratliff and Murray, 2014),
using a Ricker-Marr wavelet. (Other Gaussian-type wavelets yielded similar power spectra
and characteristic time-scales.)
The first two modes in these data are both rotational (Fig. 3b). The first, a rotation toward
the north, accounts for 51% of the observed shoreline variability with a peak time-scale at
~21 months (and a local saddle at ~12 months). The second, a rotation toward the south,
accounts for 32% (~6–7 months) and agrees closely with the $T_c$ calculated independently
from the normalised storm wave-energy flux. In previous applications of PCA to >25 years
of long-term profile data (Short and Trembanis, 2004) and 5 years of quad-bike
measurements (Harley et al., 2011a, 2015) at Narrabeen-Collaroy, rotational behaviour was
secondary (26% of shoreline variability around its mean position) to a dominant mode
(~60%) of quasi-coherent, off- and onshore sand movement within the embayment. In the
extended quad-bike dataset used here (Fig. 3a), bi-directional rotation appears to become
the predominant mode after ~2010. The third and fourth modes account for 5.4% (~10–
11 months) and 2.5% (~10–11 months) of observed shoreline variability, respectively, and
might reflect "breathing" behaviour at the fulcrum and both ends of the beach, perhaps
with influences from other sources of shoreline variability, including an ephemeral inlet
near Narrabeen Headland (Fig. 1a). Approach angles of deep-water waves associated with
different types of storm system likely control the occurrence and relative strengths of the
various modes (Harley et al., 2011a, 2015).
Although resolved in two dimensions, these shoreline behaviours nevertheless inform our
one-dimensional simplification of shoreline change (Fig. 2). The spatial analysis shows that
at each position alongshore, shoreline position is moving onshore and offshore with a few
dominant modes of sediment-transport dynamics that rework the embayed beach at
characteristic time-scales. The "closed" system of the embayment makes the beach behave
as a roughly conserved physical quantity. This means that rotation-driven shoreline change
is spatially correlated, such that one side accretes approximately as much as the other side
erodes. The spectral density of shoreline change over time at any position ($y$) is insensitive
to this spatial correlation, because the absolute value of shoreline change makes the
magnitudes at one end of the embayment approximately equal to those at the other, and
thus their power spectra quantitatively similar, in turn.

## 4. Discussion and implications

Jerolmack and Paola (2010) showed that morphodynamic turbulence will tend to "shred"
(strongly modify) input perturbations with time-scales shorter than the characteristic time-
scale of the system ($T < T_c$). Only input perturbations with time-scales $T > T_c$ are likely to
be preserved (or only weakly modified) in the output signal. The various characteristic
time-scales that we estimated for the Narrabeen-Collaroy system (Fig. 4; Table 1) suggest
that input perturbations (i.e., wave-energy events) with time-scales on the order of $T <$
$\sim 10^1$ months are subject to distortion by morphodynamic turbulence, and their effects on
shoreline change will tend to get "smeared" across a range of temporal scales in the output
signal (Fig. 4).
By extension, irregular but multi-annual forcings, such as the El Niño–Southern Oscillation
(ENSO), might have a time-scale sufficiently long enough to avoid erasure by annual
cycling (Barnard et al., 2015). The power spectra for the shoreline-change and daily-
resolution storm-wave energy flux register a peak near a time interval of ~3–4 years,
consistent with ENSO forcing. Moreover, if climate-related drivers were to increase future
forcing at the annual time-scale ($T \approx T_d$), perhaps through storm frequency or intensity or
both (Emanuel, 2013), there is potential for system resonance (Binder et al., 1995; Cadot et
al., 2003; Jerolmack and Paola, 2010) that could amplify corresponding shoreline changes.
However, the collective effect of these various and variable characteristic time-scales is to
make storm-driven perturbations difficult to isolate in sparsely sampled records of
shoreline change. If cross-shore beach recovery is rapid – that is, if most of the sediment
shifted off a beach during a storm is stored in a nearshore bar and then swept back
onshore in a matter of days to weeks afterward (Birkemeier, 1979; List et al., 2006; Phillips
et al., 2017) – then the magnitude of shoreline change driven by a storm event may appear
damped even in a monthly survey of beach position. When such large fluctuations are so
ephemeral, only high-frequency sampling can hope to capture their fullest extents (Splinter
et al., 2013; Phillips et al., 2017). And even then, nearshore beach dynamics may still
ultimately obscure the magnitude of direct environmental forcing because of the complex
transformation that offshore wave energy undergoes across the surf zone (Plant et al.,
2006; Coco et al., 2014).
Intrinsic time-scales for behavioural modes of beach change along open coastlines may be
different from those for embayed settings. Where alongshore spatial scales are large (~$10^1$–
$10^2$ km), the cumulative, diffusive effect of alongshore sediment transport is an especially
effective shredder (Lazarus et al., 2011a, 2012). Ratliff and Murray (2014) suggest the
diffusive scaling evident in their modelling results implies that characteristic time-scales
increase nonlinearly with embayment length alongshore. They list other factors that could
likewise change the characteristic time-scales, such as wave height, sediment type, and the
aspect ratio of headlands relative to the bay (which would affect local wave height through
wave shadowing). Broadly posed, where the influence of alongshore sediment transport is
significant and the beach system is "open" (rather than "closed" by headlands that make
sand a conserved quantity), then the longer the beach, the more effective the system will be
at shredding high-frequency signals. Were the same high-resolution spatio-temporal data
available for ~$10^4$ m of open sandy coastline as it is for Narrabeen-Collaroy, a comparable
analysis might highlight a series of progressively larger characteristic time-scales for
reversing erosion hotspots, alongshore sand waves, and fluctuations in alongshore
curvature (List et al., 2006; Lazarus and Murray, 2007, 2011; Lazarus et al., 2011a, 2012).
Signal shredding may be strongest when coupled to human manipulations of natural
shoreline behaviour (McNamara and Werner, 2008a, 2008b; Williams et al., 2013; Lazarus
et al., 2011b; Lazarus et al., 2016).
In an ideal source-to-sink sedimentary system with perfect storage, output flux would be
faithfully recorded in the sink stratigraphy. The majority of work in morphodynamic
turbulence and signal shredding comes from efforts to puzzle out what information
stratigraphic records do and do not convey about environmental forcing (Paola et al.,
2018). For beach systems, that may mean large forcing events like major coastal storms,
even when we can record their effects, probably tell us less about the dynamics of long-
term shoreline change – and vice versa – than we would wish to know. Empirical evidence
of signal shredding in the shoreline-position data from the Narrabeen-Collaroy system
demonstrates how, and suggests why, signatures of individual storm impacts can be
obscured or erased in long-term observational records, even those recorded at a reasonably
high temporal resolution. Jerolmack and Paola (2010) recommend using controlled
experiments to gain vital mechanistic insight into morphodynamic turbulence. Here, the
effects of system size, input flux, the magnitudes of major disturbance events and potential
resonant amplification ($T \approx T_c$) could be tested systematically across a broad parameter
space for coastal systems. In exploring the dynamics of signal shredding, controlled
experiments would also illuminate characteristic time-scales for fundamental processes of
sediment transport in coastal environments.

## Acknowledgements

EDL thanks A. Ashton and D. McNamara for discussions about signal shredding in
shoreline data, dating back to the publication of Jerolmack and Paola (2010). This work
was supported by funding (to EDL) from the NERC BLUEcoast project
(NE/N015665/2) and a University of Southampton Global Partnerships Award. Since
2004, the ongoing beach monitoring program at Narrabeen-Collaroy has been funded by
the Australian Research Council (Discovery and Linkage), Warringah and Northern
Beaches Councils, NSW Office of Environment and Heritage (OEH), SIMS foundation,
and the UNSW Faculty of Engineering (see Turner et al., 2016). We are grateful to K.
Ratliff, A. Ashton, and an anonymous reviewer for constructive comments that improved
the manuscript.

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

**Figures, Tables, and Captions**

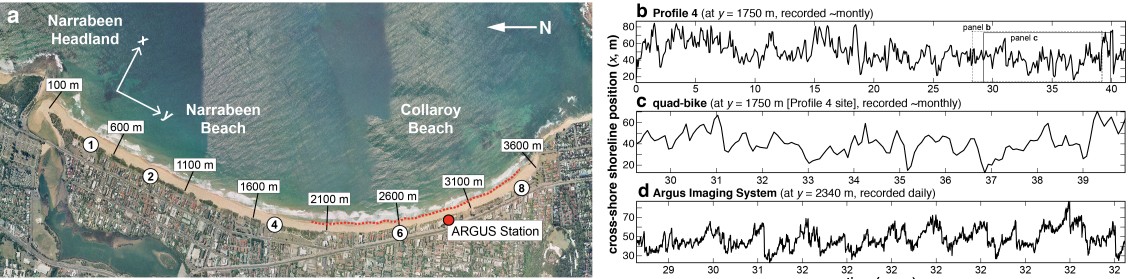


**Figure 1. (a)** Narrabeen-Collaroy beach, with locations of long time-series profiles and
Argus Imaging System coverage. Alongshore coordinates (*y*) are relative to the northern
end, below Narrabeen Headland. **(b)** Long-term time-series of cross-shore shoreline
position (0.7 m contour) at Profile 4, measured approximately monthly between 1976–
2017. Time axis is in years since first measurement (27 April 1976). **(c)** Time-series of
cross-shore shoreline position at alongshore location *y* = 1750 m (aligned with Profile 4),
measured by quad bike approximately monthly between 2005–2017. **(d)** Time-series of
cross-shore shoreline position at alongshore location *y* = 2340 m, measured daily by an
Argus Imaging System between 2005–2016. Boxes (dotted, solid) in panel (b) frame the
temporal coverages for the time-series in panels (c) and (d).

**shoreline-change analysis (output signal)**

**storm-wave analysis (input signal)**

**Figure 2.** *Shoreline-change analysis (upper panels):* Alongshore median of the absolute value of monthly shoreline change from **(a)** long-term Profiles 1, 2, 4, 6, and 8, **(b)** monthly shoreline position from the RTK-GPS quad-bike surveys, and **(c)** a 850 m reach of the Argus coverage ($y$ = 1950–2800 m). **(d)** Wavelet-derived power spectra for the three shoreline-change signals, respectively, showing a transition from non-stationary to stationary at time-scales ~$10^1$ mos. A power function fitted to the three spectra, combined, for scales up to ~12 mos, returns a scaling exponent = 0.66. *Storm-wave analysis (lower panels):* **(e)** Monthly and **(f)** daily total storm wave-energy flux between 2005–2017 (normalised to their respective maxima), used here to represent forcing input. **(g)** Power spectra for the storm-wave energy flux in (e) and (f). Labelled circles emphasise major peaks in spectral density at various time scales. Grey bar in (d) and (g) indicates an estimated characteristic time-scale $T_c$ = 4–6 months, based on normalised beach width relative to mean normalised wave-energy forcing.

540

541

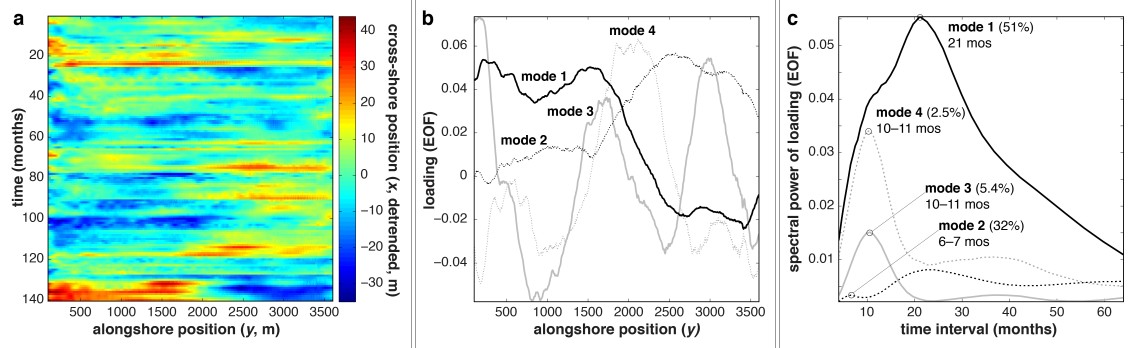

Figure 3. (a) Detrended (in time) shoreline position, measured approximately monthly by
quad bike, with north at left (corresponding to Fig. 1a). (b) Orthogonal PCA modes,
representing variance about the mean shoreline position, and (c) wavelet-derived power
spectra of each mode, where the first local maximum indicates the characteristic time-scale
for that mode.

547

548

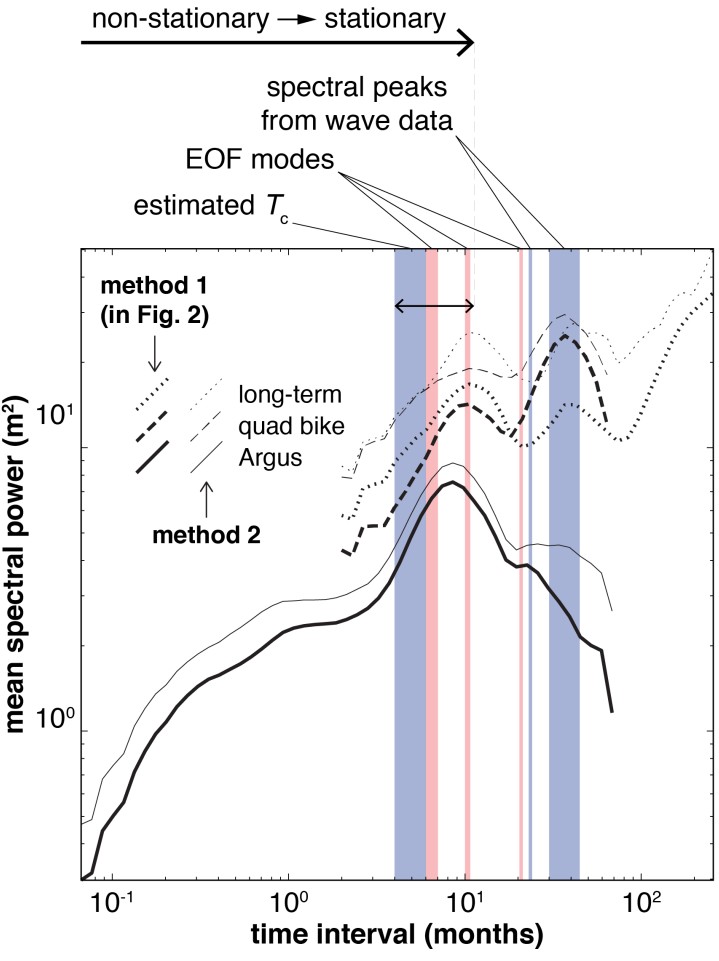

549

**Figure 4.** Compilation of power spectra from shoreline-change data in relation to different
characteristic time-scales for environmental forcing (blue/dark bars) and intrinsic physical
processes (red/light bars). Thick black lines indicate power spectra shown in Fig. 2d,
derived from the alongshore median absolute value of shoreline change through time
("method 1"). Thin grey lines show the median spectral densities of power spectra of
shoreline change through time (detrended, absolute value) at each position alongshore for
the three survey types ("method 2"), shown in Fig. S1. We plot them together here to
demonstrate their comparability. Double-ended arrow indicates transition zone in the
spectral density from non-stationary to stationary by a temporal interval on the order of
$\sim 10^1$ months.

560

561 **Table 1.** Compilation of characteristic time-scales in Figs. 2 & 4.

| Data source | Characteristic time-scales (mos) |
|---|---|
| **Shoreline-change datasets** | |
| *Method 1 (alongshore median absolute value of shoreline change)* | |
| long-term profiles (monthly) | 11, 37–42 |
| quad-bike surveys (monthly) | 11, 37–42 |
| Argus system (daily) | ~1, 9, 23 |
| | |
| *Method 2 (median spectral power of absolute value of shoreline change over time at each position alongshore)* | |
| long-term profiles (monthly) | 11–12, 42, 56 |
| quad-bike surveys (monthly) | 12, 37–42 |
| Argus system (daily) | ~1, 8–10, 26, 34 |
| | |
| **Storm-wave energy forcing** | |
| estimated $T_c$ (normalised $L/E$) | 4–6 |
| storm-wave $E$ flux (monthly) | 24 |
| storm-wave $E$ flux (daily) | ~2, 30–45 |
| | |
| **EOF modes of embayed beach behaviour** | |
| Mode 1 (51%, rotational) | 12–14, 21 |
| Mode 2 (32%, rotational) | 6–7, 22–26 |
| Mode 3 (5.4%, breathing & other) | 10–11, 36–42 |
| Mode 4 (2.5%, breathing & other) | 10–11, 36–42 |

562