# Peer review of "Environmental signal shredding on sandy coastlines"

_Earth Surface Dynamics, 2018_

## Referee Comment (RC1) · K. Ratliff (Referee) · 5 Nov 2018

General Comments:
"Environmental signal shredding on sandy coastlines" by Lazarus et al. extends the notion of morphodynamic turbulence, which has previously been explored in unidirectional sedimentary systems, to bidirectional changes in shoreline position over time. This work is relevant to ESurf readers and has important implications for the reconstruction of historic shoreline change, as well as for future long-term shoreline change predictions. The paper is well-written and clearly presented. My only significant question pertains to the sole use of the storm waves as an input flux, rather than the entire wave energy flux time series, when comparing to the daily (Argus) shoreline change dataset (see 'Specific Comments' below).

[Figure]

Specific Comments:

- l. 26: perhaps caveat that the "large forcing events" that are shredded are relatively short-term, i.e., those operating on < months time scales

- l. 131-133: The authors use the storm wave threshold (95th percentile of deep-water significant wave height) as a cutoff for the input flux time series. Particularly for the Argus dataset, it would be interesting and informative to compare the daily wave energy flux in its entirety to the shoreline change information (rather than just the storms). Are seasonal time scales evident in the daily wave energy flux time series? If so, this could be an interesting point of comparison (vs. the storm wave energy flux) that could strengthen the discussion of signal perseverance at longer time scales. An additional figure/histogram plotting the distribution of total wave energy for binned wave heights at Narrabeen-Collaroy beach from 2005-2017 could also be useful to illustrate the relative influence of storm waves.

- l. 138: include reference

- l. 226: "if climate-related drivers were to increase future forcing at the seasonal time-scale" – give an example of this – increasing storminess? Perhaps include a reference.

- l. 292-294: Re-word last sentence, as it does not read clearly.

- Figure 3B: Consider shortening x-axis to match 3A – I assume it's the same range as the Argus dataset, but the mismatch in x-scales between A and B and the data gap could be misleading and/or confusing.

Technical Corrections:

- l. 172: correct spelling of "Ratliff" in reference (no 'e')

- l. 204: should reference figure 4B, not C

---

## Referee Comment (RC2) · Anonymous Referee #2 · 6 Nov 2018

In the manuscript "Environmental signal shredding on sandy coastlines", the authors investigate if signals of external forcing can be deciphered from the record of shore-line position. They find evidence of morphodynamic turbulence in a sandy shoreline system, which has implications for both interpreting the record of shoreline change and what information may be extracted from repeat surveys at different scales. The authors should be commended for this investigation of signal processing of coastline data. This study is a unique contribution to the scientific community, with important new insights on interpreting coastline change. I recommend that the manuscript be accepted for publication. Below I detail some unanswered questions about data or assumptions, missing pieces of information, and broader questions for the authors.

1. Explain why using the alongshore average of shoreline change is representative of

the system (Fig. 2). It's unclear that it is the best metric to use for this spectral analysis, especially with rotational modes being the two primary contributors to shoreline change (Fig.4).

2. What is the source of the data that was used to create figure 3? Should be added to "Setting and dataset" section. Is it from one source or multiple?

3. L143-148: Elaborate on the physical interpretation of the wave energy flux being unorganized and stationary in contrast to the transitional shoreline change. It seems there is a lack of intrinsic characteristic timescale, but why is that important to include?

4. Consider using your wave data (heights and periods) to calculate the expected sediment flux using a variation of the CERC formula. How would this Qs signal compare to the signal produced by +/- values of shoreline change? Since the modes are rotational, it may be informative to use if AST is the majority of sediment fluxes. The modes of shoreline change by Ratliff and Murray suggest that there should be a causative link.

5. L 204 – 213: The authors describe the first two modes as rotational and detail how much change each mode accounts for, but do not describe the types of modes the third and fourth are. Though they represent little change, it would be helpful to include the mode. The magnitudes from the PCA analysis would also be helpful to include.

6. In regime 1 of the spectral power plots in Figure 2 (where the spectral power is a power law function of time scale), is the slope of the power law curve meaningful? Is it different between the data sets? There should be more details in the text of what is/can be quantified out of the power law to lead to the interpretation of morphodynamic turbulence.

7. In Figure 2, the arrows are not explained. I think they are showing that the data from one plot goes into making the next, but it appeared at first that they were pointing to something unexplained. Please clarify the purpose of the arrows. Please clarify the type of data used for each plot. It's difficult to discern as is. Could also be clearer

by labeling the columns. And it is easy to miss the timescales. A separate plot for timescales would make the connection clearer.

8. Line 247 has a typo – extra "and" in the sentence.

---

## Short Comment (SC1) · 9 Nov 2018

This manuscript submitted by Lazarus et al. covers and interesting subject matter. I had the opportunity to read through the in-discussion manuscript, and this prompted a few questions/suggestions that I believe could contribute to the final published version. Note that these review comments do not constitute a full review.

- Andrew Ashton

L22. L23 "detailed" is used twice in a row meaning different things. Or undefined things.

L109-111. Could use more discussion of how shoreline change is converted to flux and the physical motivation behind this. Particularly here where this is first discussed.

[Figure]

Also having read through this seems to be the full methods explanation. Overall the MS could use more detailed methods. In this case, the connection between the wave flux and shoreline changes needs to be made more explicit. Right now just implied. This also needs to be separated into the cross-shore (alongshore averaged) and the alongshore-varying examples used later.

L141-2. Why using "q" for something that is not directly flux. Leading.

L147. Overall, I feel that the -> q analogy is somewhat circumspect. I wish the authors could explain/justify much better. It strike me that they are forcing their results a bit too much into the specific framework of the Jerolmack and Paola (2010) rather then just being inspired by this work. The latter makes more sense as there are functional differences between the systems.

L158. Explicitly define the criteria for beach width. This probably does not affect the results, as L is just a scaling coefficient. Maybe better to scale with max – min? My concern is that on many developed/anthropogenically affected beaches, the beach width itself is set by things like development, how dune lines are locally determined, and the location of symbolic fencing.

L195. Could reference other, older works on PCA analyses of beach signals. Just an idea, I do not have specific examples on hand.

L213. Overall I feel that a constitutive connection is missing between the analyses of Fig 2 and Fig 4. Analysis one (Fig 2) is alongshore-averaged shoreline position. Analysis two (Fig 4) is about modes of change of shoreline position about the average (or at last most modes are, such as rotation, breathing, etc.). It would be helpful add more glue to put these concepts together together.

L232. I feel that overall the rice pile analogue is overemphasized. This example is constantly forced but the field site is not. I presume Fig 3 is meant to convey that there is weak spectral forcing in the data, but that should be different than constant forcing,

no?

L246-7. Um?

L250. Add some references here.

Fig 2. Please make this figure easier to read. The whole alphabet is not needed to signify the plots (not referenced as such in the text). Would be better to give titles to the columns and find a way to make the row descriptions more obvious (not hidden in the axis text).

Fig 4. Please use color or at the very least different line types here.

Overall, I think that the MS could benefit from some form of summary plot/s. Values such as the tc for the different analyses should be summarized to make the point. Instead we are left with the spiderweb thin blue lines on the plots. On the log scale nobody can even read what these numbers are. I'm not a big fan of tables, but at the least a table of the T values would help. Even better if a plot could be figured out.

---

## Author Comment (AC1) · 14 Nov 2018

We thank Dr Ratliff for her constructive comments.

In a forthcoming revision, we will integrate these line-specific suggestions and technical corrections. (And EDL apologises for misspelling 'Ratliff' at L172.)

Regarding our use of storm waves: a more inclusive threshold is something we can pursue. Previous work (see Phillips et al., 2017) has examined short-term beach recovery at Narrabeen in detail – analysis that we can summarise in a revision. Harley et al. (2011) published wave roses for the Sydney waverider buoy representing significant wave heights and directions 1992–2010, and demonstrated a seasonal pattern in beach rotation – the storm-wave energy flux reflects seasonality to a certain extent.

Again, this is context that we can summarise in a revision.

REFERENCES:

Harley, M. D., Turner, I. L., Short, A. D., & Ranasinghe, R. (2011). A reevaluation of coastal embayment rotation: The dominance of cross‐shore versus alongshore sediment transport processes, Collaroy‐Narrabeen Beach, southeast Australia. Journal of Geophysical Research: Earth Surface, 116(F4).

Phillips, M. S., Harley, M. D., Turner, I. L., Splinter, K. D., & Cox, R. J. (2017). Shoreline recovery on wave-dominated sandy coastlines: the role of sandbar morphodynamics and nearshore wave parameters. Marine Geology, 385, 146-159.

---

## Author Comment (AC2) · 14 Nov 2018

We thank Dr Ashton for his constructive comments on the manuscript, and we agree that these suggested amendments will ultimately improve the final version.

In a revision, we will incorporate the various line edits Dr Ashton has flagged, and will elaborate on the sections he suggests are under-supported (or would otherwise benefit from further discussion/explanation).

We will also include a summary table (for easier reference) of the key time scales shown in Figs. 2–4, and revise the figures to improve their clarity.

[Figure]

2018.

---

## Author Comment (AC3) · 14 Nov 2018

We are grateful to R#2 for considered and constructive comments.

A number of these notes echo (or are echoed by) comments posted here by other readers, and we will incorporate the various recommendations into a forthcoming revision. They are all elements or aspects that we can address, either with additional analysis or by more fully describing evidence established by previous work on the Narrabeen system.

---

## Author Response (AR1)

**esurf-2018-72** (Lazarus *et al.*) – RESPONSE TO REVIEWERS

December 2018

Dear Editors –

Thank you for the opportunity to revise this submission. Following the constructive recommendations from our reviewers, we have made substantial changes to the manuscript, which we detail below.

We show reviewer comments in *italics*; our replies are in **bold**; and excerpts of new text are coloured blue.

Thank you for your continued time and consideration. We look forward to further correspondence.

Kind regards,

EDL (*et alia*)
* * *
**Reviewer #1 (K Ratliff)**

*L26: perhaps caveat that the "large forcing events" that are shredded are relatively short-term, i.e., those operating on < months time scales*

**Amended sentence now reads (L26–27):**

"This suggests that the physical effects of annual (or intra-annual) forcing events, including major storms, may convey less about the dynamics of long-term shoreline change…"

*L131-133: The authors use the storm wave threshold (95th percentile of deep-water significant wave height) as a cutoff for the input flux time series. Particularly for the Argus dataset, it would be interesting and informative to compare the daily wave energy flux in its entirety to the shoreline change information (rather than just the storms). Are seasonal time scales evident in the daily wave energy flux time series? If so, this could be an interesting point of comparison (vs. the storm wave energy flux) that could strengthen the discussion of signal perseverance at longer time scales. An additional figure/histogram plotting the distribution of total wave energy for binned wave heights at Narrabeen-Collaroy beach from 2005-2017 could also be useful to illustrate the relative influence of storm waves.*

**Much of the wave-forcing analysis that R#1 suggests has been published previously. To clarify and explain our use of storm waves (rather than a broader range of waves), we have revised this section to now read (L146–152):**

"Previous work on Narrabeen-Collaroy has demonstrated that the relationship between wave-energy flux and shoreline change is strongest for storm waves (Harley et al., 2009; Phillips et al., 2017). By isolating storm waves, we do not mean to suggest that lower-energy waves do not move sediment. However, changes in nearshore bar and beach morphology tend to emerge far more slowly than the high-frequency variability of low-energy wave forcing (Plant et al., 2006), and, in this case, we are interested in the conditions under which an input flux could be preserved in the shoreline response signal. We defined storm wave conditions by a threshold corresponding to the 95th percentile of deep-water significant wave height…"

*L138: include reference*

**We now cite Herbich (2000) (L159).**

*l. 226: "if climate-related drivers were to increase future forcing at the seasonal time-scale" – give an example of this – increasing storminess? Perhaps include a reference.*

**Now citing Emanuel (2013), we have amended this sentence to read (L283–286):**

"Moreover, if climate-related drivers were to increase future forcing at the annual time-scale ($T \approx T_c$), perhaps through storm frequency or intensity or both (Emanuel, 2013), there is potential for system resonance (Binder et al., 1995; Cadot et al., 2003; Jerolmack and Paola, 2010) that could amplify corresponding shoreline changes."

*L292-294: Re-word last sentence, as it does not read clearly.*

**Revised to now read (L332–334):**

"In exploring the dynamics of signal shredding, controlled experiments would also illuminate characteristic time-scales for fundamental processes of sediment transport in coastal environments."

*Figure 3B: Consider shortening x-axis to match 3A – I assume it's the same range as the Argus dataset, but the mismatch in x-scales between A and B and the data gap could be misleading and/or confusing.*

**In response to this and related comments, we have made an entirely new Fig. 2 – which absorbed the wave-related plots shown in the original Fig. 3. We have fixed offsets in plots of power spectra and scale ranges that previously complicated direct comparisons.**

*L172: correct spelling of "Ratliff" in reference (no 'e')*

**We have corrected the citation – with apologies for the typo.**

*l. 204: should reference figure 4B, not C*

**Corrected as noted (now in Fig. 3).**

**Reviewer #2**

*1. Explain why using the alongshore average of shoreline change is representative of the system (Fig. 2). It's unclear that it is the best metric to use for this spectral analysis, especially with rotational modes being the two primary contributors to shoreline change (Fig. 4).*

**Instead of using mean alongshore position to calculate shoreline change, we now present results based on the median absolute value of shoreline change at each position alongshore. Qualitatively, our results are the same (and perhaps even**

cleaner). We also show the results of an alternative method, which takes the median of the power spectra from the absolute value of shoreline change at each position alongshore (in each of the three datasets). We present those ancillary results in a new pair of supplemental figures (Figs. S1 & S2), which also compare a wavelet-derived spectral analysis with a Fast Fourier Transform.

And, to explain how this treatment of the shoreline data dovetails with the embayed-beach dynamics discussed later in the manuscript (in Section 3.3), we have added the following (L258–268):

"Although resolved in two dimensions, these shoreline behaviours nevertheless inform our one-dimensional simplification of shoreline change (Fig. 2). The spatial analysis shows that at each position alongshore, shoreline position is moving onshore and offshore with a few dominant modes of sediment-transport dynamics that rework the embayed beach at characteristic time-scales. The "closed" system of the embayment makes the beach behave as a roughly conserved physical quantity. This means that rotation-driven shoreline change is spatially correlated, such that one side accretes approximately as much as the other side erodes. The spectral density of shoreline change over time at any position ($y$) is insensitive to this spatial correlation, because the absolute value of shoreline change makes the magnitudes at one end of the embayment approximately equal to those at the other, and thus their power spectra quantitatively similar, in turn."

*2. What is the source of the data that was used to create figure 3? Should be added to "Setting and dataset" section. Is it from one source or multiple?*

We now clarify that the wave data come from the Sydney waverider buoy:

"We also used deep-water wave data compiled from hourly records logged between 2005–2017 by the Sydney waverider buoy, located approximately 11 km offshore of the study area."

*3. L143-148: Elaborate on the physical interpretation of the wave energy flux being unorganized and stationary in contrast to the transitional shoreline change. It seems there is a lack of intrinsic characteristic timescale, but why is that important to include?*

We have significantly reworked this part of the manuscript (Section 3.1) to improve its clarity – not only by revising the text, but also by replacing the original Figs. 2 & 3 with a new Fig. 2.

The interpretation of the input signal now reads (L160–173):

"We calculated monthly and daily total storm-wave energy fluxes corresponding to the monthly and daily shoreline time-series (Fig. 2e,f), and transformed them into power spectra to demonstrate that the forcing (input) and response (output) spectra are not the same (Fig. 2d,g). Where the spectral density of shoreline change is non-stationary (correlated) over a range of relatively short time-scales (Fig. 2d), the spectral density of wave forcing is comparatively stationary (uncorrelated) over the same range (Fig. 2g). The monthly wave-energy time-series shows a peak in spectral density at ~24 mos, but with no clear comparator in the shoreline-change spectra. The daily wave-energy spectrum rises at the long-interval end of its range to a broad peak at ~30–45 mos (Fig.

2g), which overlaps with a local maximum in the shoreline-change spectra at ~37–42 mos (Fig. 2d).

Even in this one-dimensional representation, the sediment-transport processes of shoreline change have transformed an input signal into a quantitatively distinct output signal. To place these input/output spectral patterns in the context of physical processes that might explain them, we explored characteristic time-scales of key embayed-beach dynamics."

*4. Consider using your wave data (heights and periods) to calculate the expected sedi- ment flux using a variation of the CERC formula. How would this Qs signal compare to the signal produced by +/- values of shoreline change? Since the modes are rotational, it may be informative to use if AST is the majority of sediment fluxes. The modes of shoreline change by Ratliff and Murray suggest that there should be a causative link.*

**We do not pursue a calculation of expected sediment flux in this revision because we no longer call the output signal a flux (L105–108):**

"In our beach example, rather than considering sediment flux directly, we tracked the change in shoreline position, $d_x$ (in m), between consecutive time steps at a given position alongshore ($y$)."

**A comment by A Ashton (below) makes a related suggestion.**

**Furthermore, we have tried – such as at L258–268, excerpted above (see R#2's first comment) – to better explain how patterns in the one-dimensional and two-dimensional representations of shoreline behaviour are related.**

*5. L 204 – 213: The authors describe the first two modes as rotational and detail how much change each mode accounts for, but do not describe the types of modes the third and fourth are. Though they represent little change, it would be helpful to include the mode. The magnitudes from the PCA analysis would also be helpful to include.*

**We now include this information in the text (L251–255) and in Fig. 3c.**

*6. In regime 1 of the spectral power plots in Figure 2 (where the spectral power is a power law function of time scale), is the slope of the power law curve meaningful? Is it different between the data sets? There should be more details in the text of what is/can be quantified out of the power law to lead to the interpretation of morphodynamic turbulence.*

**Our new Fig. 2 shows the power spectra from all three shoreline-position datasets relative to each other. We now include a slope of the non-stationary (correlated) reach of the power spectra, along with the following interpretation (L128–136):**

"Like the sedimentary systems described by Jerolmack and Paola (2010), the spectral density of the one-dimensional shoreline-change term $d_x(t)$ yields a pattern with two regimes (Fig. 2d). A non-stationary regime extends over shorter time-scales, such that spectral density versus time-scale are correlated by a power law. This relationship transitions at ~9–11 mos into a comparatively stationary (uncorrelated) regime over longer intervals. (A power function fitted to the three spectra, combined, for scales up to ~12 mos, returns a scaling exponent = 0.66, but the physical significance this slope value remains unclear.) This two-regime pattern in the power spectrum (Jerolmack and Paola,

2010) serves as an initial indication that signal shredding may be inherent in the dynamics of sandy beach systems."

*7. In Figure 2, the arrows are not explained. I think they are showing that the data from one plot goes into making the next, but it appeared at first that they were pointing to something unexplained. Please clarify the purpose of the arrows. Please clarify the type of data used for each plot. It's difficult to discern as is. Could also be clearer by labeling the columns. And it is easy to miss the timescales. A separate plot for timescales would make the connection clearer.*

**Again, we have addressed these elements by constructing a completely new figure.**

*8. Line 247 has a typo – extra "and" in the sentence.*

**Fixed, as suggested.**

**Commenter #3 (A Ashton)**

*L22. L23 "detailed" is used twice in a row meaning different things. Or undefined things.*

**Amended second use of "detailed" so that L21–22 now read:**

"…shorelines retain almost no detailed information about their own past positions. Here, we use a high-frequency, multi-decadal observational record of shoreline position…"

*L109-111. Could use more discussion of how shoreline change is converted to flux and the physical motivation behind this. Particularly here where this is first discussed. Also having read through this seems to be the full methods explanation. Overall the MS could use more detailed methods. In this case, the connection between the wave flux and shoreline changes needs to be made more explicit. Right now just implied. This also needs to be separated into the cross-shore (alongshore averaged) and the alongshore-varying examples used later.*

**We have used this comment to motivate significant revisions throughout Sections 3.1 & 3.3. We have expanded our methodological explanations wherever they appear, and we have made a particular effort to better link the shoreline-change and wave-energy time-series that we treat as system output and input, respectively.**

*L141-2. Why using "q" for something that is not directly flux. Leading.*

*L147. Overall, I feel that the -> q analogy is somewhat circumspect. I wish the authors could explain/justify much better. It strike me that they are forcing their results a bit too much into the specific framework of the Jerolmack and Paola (2010) rather then just being inspired by this work. The latter makes more sense as there are functional differences between the systems.*

**As we note above, to R#2, we no longer refer to the shoreline-change signal as a flux (L105–108).**

**Moreover, we have revised the bulk of this submission according to the spirit of this comment – that we be inspired by Jerolmack & Paola (2010) rather be strict**

adherents to its template. Even in the coauthors' independent readings of this revision, the shift in presentation has improved the manuscript fundamentally.

*L158. Explicitly define the criteria for beach width. This probably does not affect the results, as L is just a scaling coefficient. Maybe better to scale with max − min? My concern is that on many developed/anthropogenically affected beaches, the beach width itself is set by things like development, how dune lines are locally determined, and the location of symbolic fencing.*

**Indeed, because we normalize $L$, the definition of beach width does not affect the results. (In a previous draft, we used a definition of $max − min$, but ultimately opted to describe the "full" beach width captured by the cross-shore profile.) However, for clarity, we have amended this line to read (L183–186):**

"We assume that the system size $L$ is equivalent to maximum cross-shore beach width, defined here as the cross-shore distance from a fixed landward reference point to mean sea-level (Harley and Turner, 2008; Harley et al., 2011b)."

*L195. Could reference other, older works on PCA analyses of beach signals. Just an idea, I do not have specific examples on hand.*

**We now refer (L231–232) to Winant et al., 1975; Aubrey, 1979; Clarke and Eliot, 1982; Hsu et al., 1994; Dail et al., 2000; and Short and Trembanis (2004).**

*L213. Overall I feel that a constitutive connection is missing between the analyses of Fig 2 and Fig 4. Analysis one (Fig 2) is alongshore-averaged shoreline position. Analysis two (Fig 4) is about modes of change of shoreline position about the average (or at last most modes are, such as rotation, breathing, etc.). It would be helpful add more glue to put these concepts together together.*

**We have worked to address this disconnect – which R#2 also raises – throughout Section 3 of the text, and by presenting entirely new Figs. 2 & 4 (where the latter is a synthesis figure, as recommended below).**

*L232. I feel that overall the rice pile analogue is overemphasized. This example is constantly forced but the field site is not. I presume Fig 3 is meant to convey that there is weak spectral forcing in the data, but that should be different than constant forcing, no?*

**We have removed this discussion of the rice-pile analogue from the text (in keeping with the "inspired by" comment, above), and have clarified (with the new Fig. 2) our presentation and interpretation of input forcing.**

L246-7. Um?

**Revised, as suggested.**

*L250. Add some references here.*

**For the clause "… if most of the sediment shifted off a beach during a storm is stored in a nearshore bar and then swept back onshore in a matter of days to weeks afterward…" (L290–292), we now include Birkemeier (1979), List et al. (2006), and Phillips et al. (2017).**

*Fig 2. Please make this figure easier to read. The whole alphabet is not needed to signify the plots (not referenced as such in the text). Would be better to give titles to the columns and find a way to make the row descriptions more obvious (not hidden in the axis text).*

**Addressed, with an entirely new Fig. 2.**

*Fig 4. Please use color or at the very least different line types here.*

**Revised, as suggested (now Fig. 3).**

*Overall, I think that the MS could benefit from some form of summary plot/s. Values such as the tc for the different analyses should be summarized to make the point. Instead we are left with the spiderweb thin blue lines on the plots. On the log scale nobody can even read what these numbers are. I'm not a big fan of tables, but at the least a table of the T values would help. Even better if a plot could be figured out.*

**Addressed with an entirely new Fig. 2,and with the addition of a new Fig. 4 as a "synthesis" plot and a new Table 1 listing all of the characteristic time-scales across the various input/output time-series and analyses.**

[revised manuscript text omitted]